# New Insights into the Microbial Profiles of Infected Root Canals in Traumatized Teeth

**DOI:** 10.3390/jcm9123877

**Published:** 2020-11-28

**Authors:** Lokeshwaran Manoharan, Malin Brundin, Olena Rakhimova, Luis Chávez de Paz, Nelly Romani Vestman

**Affiliations:** 1National Bioinformatics Infrastructure Sweden (NBIS), Lund University, 22362 Lund, Sweden; lokeshwaran.manoharan@nbis.se; 2Department of Endodontics, Umeå University, 90187 Umeå, Sweden; malin.brundin@umu.se; 3Department of Odontology, Umeå University, 90187 Umeå, Sweden; olena.rakhimova@umu.se; 4Private Practice, Stockholm, 13933 Stockholm, Sweden; luis.chavez.de.paz@gmail.com; 5Department of Endodontics, County Council of Västerbotten, 90189 Umeå, Sweden; 6Wallenberg Centre for Molecular Medicine, Umeå University, 90187 Umeå, Sweden

**Keywords:** dental trauma, pulp necrosis, root canal microbiota, pulp regeneration, endodontic pathogens

## Abstract

Traumatic dental injuries in young individuals are often exposed to the invasion of oral microorganisms that leads to pulp necrosis. Infective necrosis in permanent teeth not-fully-developed causes aberrant root formation. Regeneration endodontic treatments (RETs) have shown promising results by promoting continued root development by stem cells. Critical to the success of RET is the thorough disinfection of the pulpal space. To establish effective antimicrobial protocols for root canal disinfection, the invading microorganisms need to be identified. In the present study, we use a combination of culture-based and high-throughput molecular sequencing techniques to investigate the microbial profiles from traumatized teeth (30 cases) and controls, i.e., teeth with pulp infections not caused by trauma (32 cases). Overall, a high microbial diversity in traumatized necrotic teeth was observed. *Eubacterium yurii subsps. yurii* and margaretiae, as well as key ‘bridging oral species’ *F. nucleatum sp., Polymorphum* and *Corynebacterium matruchotti*, were highly associated with traumatized teeth. The microbial compositions of traumatized teeth differed considerably from those of infected teeth not caused by trauma. Age and tooth position also influence microbial compositions. In conclusion, we show that the root canal microflora of traumatized teeth is highly diverse, and it differs from root canal infections not caused by trauma.

## 1. Introduction

Dental trauma injuries during childhood and adolescence can have an adverse impact on oral health throughout life. Population-based studies estimate that the worldwide prevalence of dental trauma in permanent young teeth is between 16–40% [1]. Additionally, the incidence rate for dental trauma indicates an upward trend (incidence per 1000: 18.9 in 2011 to 28.5 in 2013) [2]. After traumatic injuries to teeth, dental pulp and neurovascular supply can be disrupted, and microorganisms may invade the compromised pulp tissue, initiating an infection and periapical inflammation [3]. Indeed, development of pulp necrosis occurs in up to 27% of permanent teeth with traumatic injuries, leading to serious consequences such as aberrant root formation and questionable long-term tooth survival [4].

Clinical treatment of trauma-induced apical periodontitis in teeth not-fully developed represents a challenging clinical situation. The main goal with these clinical procedures is to promote root development [5] and to prevent traumatized teeth becoming susceptible to cervical root fractures [6]. Conventional treatments are focused on the disinfection of the pulpal space systems and the closure of the open apex (apexification). Apexification procedures consist mainly of the induction of a calcific barrier in the apex, and can be achieved by the long-term application of a calcium hydroxide (CH) paste [7] or by applying a mineral trioxide aggregate cement (MTA) [8]. Nowadays, an additional therapeutic option is available: regenerative endodontic treatment (RET). RET has shown very promising results as it is a tissue regenerative procedure that promotes continued root development by stem cells [9,10]. However, one of the main challenges in RET is the persistence of microorganisms in the pulpal space, even after disinfection regimens [11], a fact that may have adverse effects in stem cells function [12]. Hence, knowledge of the microbiota profiles of traumatized teeth can be essential for treatment success.

The oral cavity provides a continuous source of bacteria, some of which, if given the opportunity, will invade the root canal system. For healthy adults, different factors such as, age, gender, diet, niches sites, and extreme environment have influenced the oral microbiome [13]. The root canal has a unique and selective anaerobic environment, which modulates synergic and antagonistic activity of colonizing bacteria, and has been widely investigated, especially in relation to primary and secondary infections [14]. However, to date, no studies have addressed the impact of environmental factors, such as dental trauma, which can provide critical insights into the root canal microbial composition.

Previous studies, using both traditional culture techniques and molecular techniques of identification such as Polymerase Chain Reaction (PCR), have shown that the microflora of infected traumatized teeth is similar to that of primarily infected permanent teeth [15]. In this study, we applied a next-generation sequencing technique to investigate the microbial flora of traumatic necrotic teeth. The aim of this study was to evaluate the microbial profile of traumatized necrotic teeth, and to compare eventual differences in microbial composition with primary infected teeth not caused by trauma.

## 2. Material and Methods

### 2.1. Study Population

Sixty-two patients presenting pulp necrosis followed by dental trauma (*n* = 30) or primary infections not due to trauma (*n* = 32) were included in this study. All samples were collected from patients aged ≥6 years (mean 34.4) and who were referred to the Department of Endodontics, Umeå University Hospital, Sweden. Patients were included if they presented untreated teeth with a diagnosis of pulp necrosis, and an adequate coronal restoration without signs of caries. Pulp necrosis was verified by cold pulp sensibility test (Endo Ice; Roeko, Langenau, Germany), electric test (Pulp Tester; Analytic Technology, Redmond, WA, USA), and radiographic evidence of periapical lesion. Inclusion criteria consisted of primary infected teeth with adequate coronal restoration, excluding teeth presenting extensive crown destruction allowing contamination from the oral cavity. Patients with advanced periodontal lesions and who received antibiotic treatment during the previous three months were excluded. The traumatized teeth had various trauma diagnoses, including hard tissue and luxation injuries. The time lapse between trauma and the referral to the endodontic department varied from 28 days up to one year and 11 months. The routes of contamination for the non-traumatized teeth were those other than active caries, including previous orthodontic treatment, mild periodontal disease, and mostly bacterial invasion due to operative procedures, such as deep restoration or crown preparation. None of the patients had clinical swellings or severe pain. Data on age, tooth position, gender, and clinical findings such as presence of sinus tract were collected at the clinical examination. The study was conducted in accordance with ethical principles of the 64th WMA Declaration of Helsinki. Ethical approval was obtained by the Regional Ethical Board at Umeå University, Sweden, with registration number Dnr (2016/520-31) for children and Dnr (2014/357-31) for the adults. All patients consented to inclusion in the study. Older children and parents signed informed consent to participate.

### 2.2. Sample Collection, Processing, and Characterization of Isolates

Samples were collected on teeth isolated with a rubber dam and using strict aseptic techniques. The tooth surface was cleaned with 30% hydrogen peroxide for one minute and NaOCl 3% for two minutes. The operation field (tooth, rubber dam, and clamp) was swabbed with 5% iodine tincture. To control the sterility of the operation field, samples of the disinfected tooth crown were taken with two foam pellets (Disposable Mini-Sponge Applicator; 3M ESPE, St. Paul, MN, USA) damped in thiosulfate solution (5%; surface control). One pellet was placed in fluid thioglycollate medium supplemented with agar (FTM). If growth occurred after seven days of aerobic incubation, the sample was excluded from the study. The other pellets were stored at −80 °C in Tris-EDTA (TE-buffer, Sigma, St. Louis, MO 63178, USA) for further DNA analysis.

For root canal samples, the access cavity was prepared with a sterile carbide bur, canals were gently filed with K-files (Dentsply Sirona Endodontics), and filled using a syringe containing sterile saline solution. The contents of the root canal were absorbed into sterile paper points and transferred to FTM. The paper points were moved to TE-buffer (1 mL) and ten-fold serial dilutions (0–10^4^) were cultured on fastidious anaerobic agar (FAA, Svenska LABFAB, #ACU-7531A) in an anaerobic atmosphere (5% CO_2_, 10% H_2_, 85% N_2_, 37 °C) for one week. Colony-forming units (CFU) were counted, and colonies with different phenotypic patterns were selected from each plate for bacterial typing. Two colonies of each phenotypic pattern were collected. The remaining sample (800 µL) containing the paper points was stored at −80 °C for DNA extraction. The experimental workflow is presented in Figure 1. Up to 10 isolates with different phenotypic patterns were selected from each plate, amplified by PCR, and sequenced to identify bacterial species as previously described [13]. Thus, aliquots of the 16S rDNA PCR products were purified and sequenced with Sanger (Eurofins MWG Operon, Ebensburg, Germany), and sequences were compared with the eHOMD database (Expanded Human Oral Microbiome Database, Forsyth, (http://www.ehomd.org)).

### 2.3. Illumina MiSeq Sequencing

DNA purification of surface controls and root canal samples were performed as previously described [16]. Illumina MiSeq sequencing was applied according to the HOMINGS protocol [17] at the Forsyth Research Institute (Cambridge, MA, USA). Briefly, the V3-V4 hypervariable regions of the 16S rDNA were PCR amplified using the forward 341F and reverse 806R primers. Amplicons were purified (AMPure beads; Beckman Coulter Genomics, Danvers, MA, USA); 100 ng of each library was pooled, gel purified, and quantified; and 20% PhiX was added to 12 pmol of the library mixture and run on the MiSeq system. Paired-end reads (2 X 250 bp) from MiSeq were then fused; barcodes, primers, and ambiguous and chimeric sequences were removed, followed by amplicon sequence variant (ASV) predictions and their respective abundances. The ASVs with total abundances of less than 10% of the samples with at least five counts were filtered from the dataset. These steps were achieved by using QIIME2 [18] in combination with DADA2 [19]. The taxonomy of the ASVs were predicted using the HOMD reference sequences PHYL [20]. Based on the taxonomy annotations, the respective HMT taxa abundances were then calculated. The taxonomy bar plots based on the relative abundances of the HMT taxa were then plotted using PHYLOSEQ [21].

### 2.4. Statistical Analysis

Differences between means for CFU/mL detected were tested using analysis of variance (ANOVA). Categorical data are presented as proportions (%), and differences between groups were tested with a Chi^2^ test. A *p*-value < 0.05 was considered statistically significant. Data handling and statistical analyses were performed using PASW Statistics 20 (IBM Corporation Route 100, Somers, New York, NY, USA).

The alpha-diversity of the samples based on the HMT taxa were then computed using PHYLOSEQ. The beta-diversity of the samples for the different subset of analysis based on the metadata were then calculated using the “Bray–Curtis” distance metric after rarefaction to even depth, which is then visualized through PCoA using PHYLOSEQ [21]. The significance of the variables from the metadata to their beta-diversity for each of the comparisons was then calculated based on PERMANOVA, using the Adonis function in the R package “vegan” [22]. To predict the HMT taxa associations to the different variables in the metadata, DESeq2 [23] based differential abundance analysis was performed for the respective variables in R. The adjusted *p*-values were then calculated using the *p*.adjust function in R. The subset of the significant taxa were then plotted using the heatmap from PHYLOSEQ [21].

## 3. Results

### 3.1. Study Population

Sixty-two teeth associated with infected root canals followed by dental trauma (*n* = 30) or primary infections not due to trauma (non-trauma) (*n* = 32) were analyzed by culture and Illumina MiSeq sequencing. Two samples were excluded due to contamination (Figure 1). Characteristics such as gender, age, tooth position, and other clinical features are presented in Table 1. Among the participants, the proportion of female and male, presence of sinus tract, and subjective rapport of pain did not differ significantly between trauma and non-trauma teeth. However, there was a significant difference between groups in age and tooth position: teeth from younger patients with anterior tooth position were most evident in the trauma group.

### 3.2. Fusobacterium Nucleatum and Slackia Exigua Were the Most Prevalent Species from Root Canals Identified by Culture

To control sterility, samples from tooth surfaces were cultured and growth was monitored seven days after incubation. As expected, no bacterial growth was obtained from tooth surfaces on almost all teeth (*n* = 60). Only two samples showed growth and were excluded from the whole study due to contamination. Bacteria was cultivable in 82% of the root canal samples (trauma = 73%; non-trauma = 90.63%). Geometric means for root canal samples were 791,396 CFU/mL, and 791,396 (8400000-20) medians (max-min) CFU/mL. Presence of mean levels of CFU/mL were approximately three times higher in the non-trauma group than in the trauma group (Table 1). However, the difference was not statistically significant.

Among the 284 isolates collected from root canals (trauma = 110; non-trauma 174), 79 different bacterial species were identified by comparing 16S rRNA gene sequence to database HOMD (Appendix A). Forty-nine species were found in samples from traumatized teeth and 59 from non-trauma teeth. An average of 5.5 different species were recovered on each tooth regarding the origin of the sample (trauma and non-trauma). *Fusobacterium nucleatum* was the most prevalent isolated species, detected in 50% and 24.14% of traumatized and non-traumatized teeth, respectively. Moreover, *F. nucleatum ssp. Polymorphum* was exclusively identified in traumatized teeth. Other cultivable common species in root canals were *Dialister invisus*, *Parvimonas micra*, *Pervotella nigrenscens*, and *Peptostreptococcus infirmun*. From 20 species exclusively detected in trauma teeth, the most common were *Alloprevotella tannerae*, *Campylobacter showae*, and *Campylobacter rectus*. Thirty species were exclusively detected in non-trauma teeth, the most prevalent being *Pseudoramibacter alactolyticus, Porphyromonas gingivalis*, and *Atopobium parvulum* (Figure 2). Notably, *Enteroccocus faecalis*, a recognized pathogen in post-treatment endodontic infections [24], was only detected in teeth not related to trauma.

Strictly anaerobic strains that were isolated from traumatized and non-traumatized teeth accounted for 81.9% and 90% of all isolates, respectively. Within strains with respiratory metabolism (aerobic, microaerophilic, and facultative anaerobic), *Capnocytophaga sp.* was only isolated in traumatized teeth, while *E. faecalis* was only cultivated in non-traumatized teeth (Figure 2, Appendix A).

### 3.3. Illumina MiSeq Sequencing Showed an Overall High Microbial Diversity

Illumina sequencing of the 16S rRNA V3-V4 amplicon libraries generated from 90 samples (surfaces = 38; root canal = 52) yielded ~13 million raw reads pairs, of which ~11 million passed quality control steps (Table 2). Sequences were classified against the eHOMD reference database containing 771 Human Microbial Taxons (HMT) and providing species-level taxonomy based on grouping 16S rRNA gene sequences at 98.5% identity [25]. As many as 551 HMTs out of 771 were presented in the samples, indicating a high diversity. The surface controls and root canal samples harbored 502 and 454 HMTs, respectively. On average, 92.2 ± 39.1 HMTs (range 29–182 HMTs) per surface control and 49.8 ± 27.8 HTMs per root canal (range 11–143) were identified.

We employed the Chao1 estimator of total richness to estimate the number of species/features per sample, and the Shannon index for community diversity [26]. Overall, Chao1 and Shannon diversity indicate a great richness of the studied bacterial communities. There was no significant difference in the alpha diversity assessed by Chao1 and Shannon indices between the samples, regardless of their origin (surface control and tooth surfaces) and trauma history (trauma and non-trauma) (Figure 3A). Further, representatives of 11 bacterial phyla (Absconditabacteria_(SR1), Actinobacetria, Bacteroidetes, Chloroflexi, Firmicutes, Fusobacteria, Gracilibacteria_(GN02), Proteobacteria, Saccharibacteria_(TM7), Spirochaetes, and Synergistetes) and 168 genera were detected in all samples. Proteobacteria and Firmicutes were the most common phyla in tooth surfaces. Of 11 bacterial phyla identified in root canal samples (trauma = 9; non-trauma = 10), five phyla (Firmicutes, Bacteroidetes, Fusobacteria, Actinobacteria, and Proteobacteria) were the most abundant regardless their association to trauma, age, gender, or tooth location (Figure 3B). Overall, the three most predominant genus presented in root canals were Fusobacterium, Prevotella, and Dialister as displayed in Figure 3C. *Fusobacterium nucleatum* was by far the most abundant HMT detected in all but nine root canal samples. For Fusobacterium, the two subspecies *F. nucleatum animalis* (HMT-420) and *F. nucleatum vincentii* (HMT-200) were among the top five most abundant HMTs in all root canal samples (Appendix A). Interestingly, the most abundant genus on tooth surfaces was Pseudomonas (Phylum: Proteobacteria, Class: Gammaproteobacteria, Family: Pseudomonadaceae, Species: *Fluorescens* HMT-612), demonstrating its DNA persistence beside strict aseptic disinfection protocol (Appendix A).

### 3.4. Trauma, Age, and Tooth Position Are Factors Influencing the Microbiota Composition

To measure differences between microbial communities, Principal coordinate analysis (PCoA) plots of the beta diversity were used and significant differences for each of the comparisons were calculated based on PERMANOVA. When this was applied on all samples in this study together, it was clear that samples recovered from surface controls clustered separately from samples from the root canals (PEMANOVA, *p* ≤ 0.001) (Figure 4A). To reveal if there was a difference between trauma and non-trauma teeth within the root canal samples, those groups were analyzed separately (Figure 4B). To elucidate the difference between trauma vs. non-trauma without the effects of age and tooth-position, community compositions were compared only with samples from the age-group of below 30 (Figure 4C), and only with samples from the frontal tooth position (Figure 4C). Moreover, it is known that the oral microbiota composition of supragingival plaque is affected by tooth position [27] and age [28,29]. In both cases, significant differences between the trauma and non-trauma samples were observed within <30-years-old group (PEMANOVA, *p* < 0.001 ****) and frontal tooth (PEMANOVA, *p* < 0.001 ****) from root canal samples. Other factors such as presence of fistula, subjective pain, and gender did not separate the samples distinctly. Moreover, when analyzing only traumatized teeth, the microbiota profile of teeth with complete root development did not group distinctly from teeth with incomplete root development. Similarly, lapse time between trauma and root canal treatment (<6 months, 6–12 months, and >12 months) did not influence the bacterial composition profile in traumatized teeth.

### 3.5. The Microbiome Composition from Root Canals Discriminates Traumatized Teeth from Other Primary Infected Non-Traumatized Teeth

As a next step, we investigated whether trauma affects the microbiome independent of age and tooth position. Thus, sub-groups of samples from patients below 30 years of age and with frontal tooth position were analyzed. There was no significant difference in the alpha diversity assessed by Chao1 and Shannon indices between these samples (Figure 5A). When we looked further into the root canal samples from this subgroup of teeth, we clearly found a high significant difference, grouping trauma teeth differently from non-trauma teeth (PEMANOVA, *p* < 0.001) (Figure 5B). Taken together, the study shows that the community composition profile of root canal samples was significantly affected by trauma.

### 3.6. Peptostreptococcaceae Yurii and Key “Bridging Species” Fusobacterium nucleatum ssp. Polymorphum and Corynebacterium Matruchotti Are Highly Associated with the Microbiota of Traumatized Teeth

The differential abundance test analysis showed a clear difference in microbiota composition between trauma and non-trauma teeth as displayed in Figure 6. Accordingly, being in the trauma group was significantly associated with higher levels of *Peptostreptococcaceae yurii* and *Selenomonas noxia*, while *Paenibacillus glucanolyticus* and *Atopobium sp._HMT_199* were associated with non-trauma teeth (Figure 6, Appendix A). Further, 18 species were associated with trauma teeth while 30 species were associated with non-trauma teeth. Notably, *F. nucleatum ssp. Polymorphum*, with a crucial role as “bridging” microorganisms [30], and *Corynebacterium matruchotti*, recently described as “hedgehog” structure of biofilm formation [31], were significantly associated with traumatized teeth.

## 4. Discussion

One of the main factors influencing the success of pulp regenerative therapies in trauma-related infections of teeth not-fully developed is the effectiveness of the antimicrobial methods used to render pulpal spaces free from bacteria. Microbiome profiling of trauma-related infections may allow the development of efficient antimicrobial methods that could be applied prior to regenerative procedures [32]. In the present study, we show that the microbiome of root canals differs in trauma and non-trauma cases. We also show that the root canal microbiome is highly diverse and is associated with species playing a crucial role in biofilm formation.

The data presented in the present study comprise the first combined culture-molecular analysis of the microflora in dental trauma cases. The introduction of culture-independent “omics” techniques has widened our knowledge on oral microbial diversity, including previously uncultivable species, uncharacterized microbial communities, and low-abundance taxa [33]. Molecular data have shown that by using only traditional culture techniques, the risk of missing keystone species [34], or even missing whole bacterial phyla, was greater. In accordance with other studies [35,36,37], this work shows that the microbiome of traumatized teeth is highly diverse and classified mainly within the phyla of *Firmicutes*, *Bacteroidetes*, *Fusobacteria*, *Actinobacteria*, and *Proteobacteria.* Contrary to other studies that found a high frequency of phyla *Protecobateria* [36], we found the phyla *Firmicutes* to be most prevalent. This discrepancy can be explained by the fact that, although not frequent in root canals samples, *Proteobacteria* was highly prevalent on tooth surfaces, suggesting the possibility of contamination with DNA regardless of strict aseptic disinfection protocol prior to sampling. The effect of the latter issue on false-positive results is of great importance and has been overlooked by many investigations applying molecular techniques to investigate root canal samples.

Previous studies analyzing the microflora in dental trauma teeth are not conclusive. Older culture-based approaches of dental trauma cases have reported a high predominance of anaerobic species belonging to the genus *Bacteroides, Coynebacterium, Peptostreptococcus*, and *Fusobacterium* [38,39]. Our culture-based results confirmed those results of early studies, where species such as *Fusobacterium nucleatum*, *Slackia exigua*, *Dialister invisus*, *Parvimonas micra*, *Pervotella nigrenscens*, and *Peptostreptococcus infirmun* were most prevalent. In a recent study by Nagata, Soares [15], a high prevalence of *Actinomyces naeslundii* (associated with failed endodontic therapy [40]), *Porphyromonas endodontalis, Parvimonas micra*, and *Fusobacterium nucleatum* was found. Our study failed to find *A. naeslundii*, but other Actinomyces species such as *A. georgiae*, *A. gerencseriae*, *A.johnsonii*, *A. lingnae*, and *A. massiliensis* were found. This apparent discrepancy can be based on the fact that previous studies have targeted only specific species.

In accordance to previous reports [41], *Fusobacterium nucleatum* was by far the most abundant phylotype associated with traumatized teeth. *F. nucleatum* is known as a key species in dental plaque formation since it expresses adhesins and serves as a “bridge” with a high number of interactions with oral bacteria [30]. Interestingly, it seems that Fusobacteria adhesion to oral pathogens is sub-species dependent, since *Streptococcus mutans* SpaP binds to RadD of *Fusobacterium nucleatum ssp. Polymorphum* but no other Fusobacteria [42]. *Corynebacterium matruchotti*, a Gram-positive pathogen highly prevalent in saliva of healthy adolescents [43], has been recently shown by CLASI-FISH combinatorial labelling and spectral imaging [31] to be a cornerstone or “hedgehog” structure in biofilm formation. *Corynebacterium matruchotti* also seems to play an important role in subgingival plaque community associated with periodontal disease, as it is found in conjunction to periodontal pathogens *Eubacterium yurii* and *Selenomonas noxia* [44,45].

Our study did not show any significant difference in root canal microbial composition between traumatized teeth with complete and incomplete root formation. These results are in agreement with a previous study reporting that the microbiota profile of traumatized teeth is similar to that of primary infected teeth [15]. Similarly, no differences were observed in the microbial profiles of traumatized teeth based on duration of infection, which was estimated by the time lapse between trauma and endodontic treatment. Interestingly, we found that the root canal microflora of traumatized teeth differed from root canal infections not caused by trauma, even when age and tooth position was considered as confounding factors. Possible reasons for these differences could be explained by micro-ecological parameters such local pH, abundance and partial pressure of oxygen, redox potential, availability of selective nutrients, and state of local host defenses [46], which can change the establishment of specific biofilm microbiota in a given niche of the oral cavity.

Besides, as described for the oral microbiome, different factors such as, age, diet, and extreme environment could have influenced the root canal diversity [13]. The number of young individuals in the non-traumatized group who present front teeth with a diagnosis of pulpal necrosis is very limited in Scandinavian countries as is reflected in our study. We investigated whether trauma affects the microbiome, independent of age, by stratifying the data into two categories (<30 years and >30 years) to reach the highest possible power. However, confirmation of the results of this study is desirable, especially the inclusion of distinct age categories that are more representative for children, young adolescents, and adults. It has been reported that the root canal microbial species recovered from asymptomatic teeth are different from those isolated from clinically symptomatic ones [47]. None of the patients in this study had clinical swellings or severe pain. Besides, when only patients with asymptomatic apical periodontitis were included in the analysis, we still find significant differences (*p* < 0.001) in the microbial compositions between the traumatized and non-traumatized groups. Although there is limited information on the exact time of bacterial colonization in the root canal, the length of time between the trauma and the endodontic treatment did not influence the composition of the microflora, which is consistent with previous results [38]. The present findings are interesting and certainly noteworthy, but further knowledge is needed to understand the biological and clinical relevance of this finding. Although the number of trauma cases available was limited, the validity of our results is improved by the robust methodology used.

## 5. Conclusions

In conclusion, our study shows that the root canal microflora of traumatized teeth differs from root canal infections not caused by trauma as assessed by illumina MiSeq sequencing. Moreover, a high microbial diversity in traumatized necrotic teeth was observed. *Eubacterium yurii* subsps. yurii and margaretiae, as well as key ‘bridging oral species’ *F. nucleatum* sp., *Polymorphum*, and *Corynebacterium matruchotti*, were highly associated with traumatized teeth.

Our study is important for the perseverance of traumatized teeth in young individuals, as it furthers our understanding of the microbial flora associated with trauma-related conditions. Furthermore, we highlight the need for more information regarding species that could be difficult to eliminate from these cases, and that could compromise the outcome of novel treatments such as RET.

## Figures and Tables

**Figure 1 jcm-09-03877-f001:**
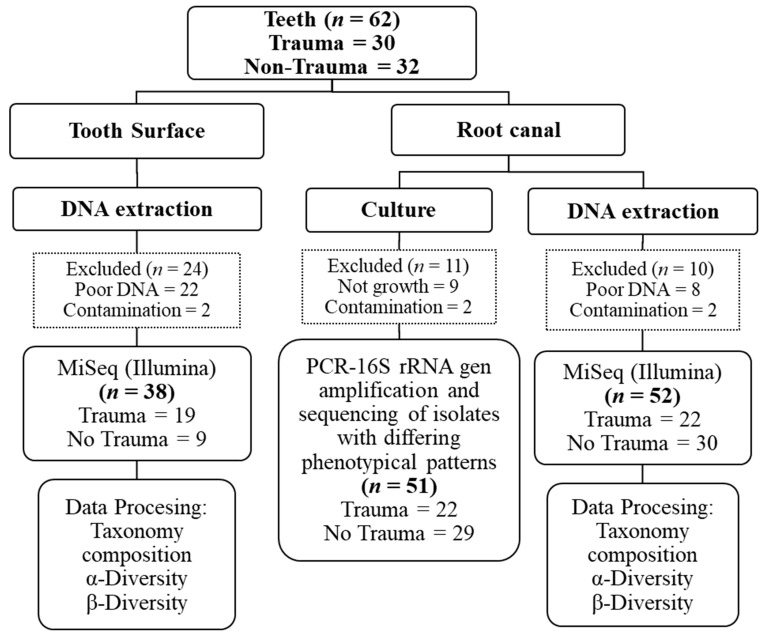
Overview of the experimental workflow. Samples were obtained from tooth surfaces and the root canal. Culture isolates were identified by Sanger sequencing of 16S rDNA PCR products and DNA was extracted from the samples for Illumina MiSeq sequencing of microbiome analysis.

**Figure 2 jcm-09-03877-f002:**
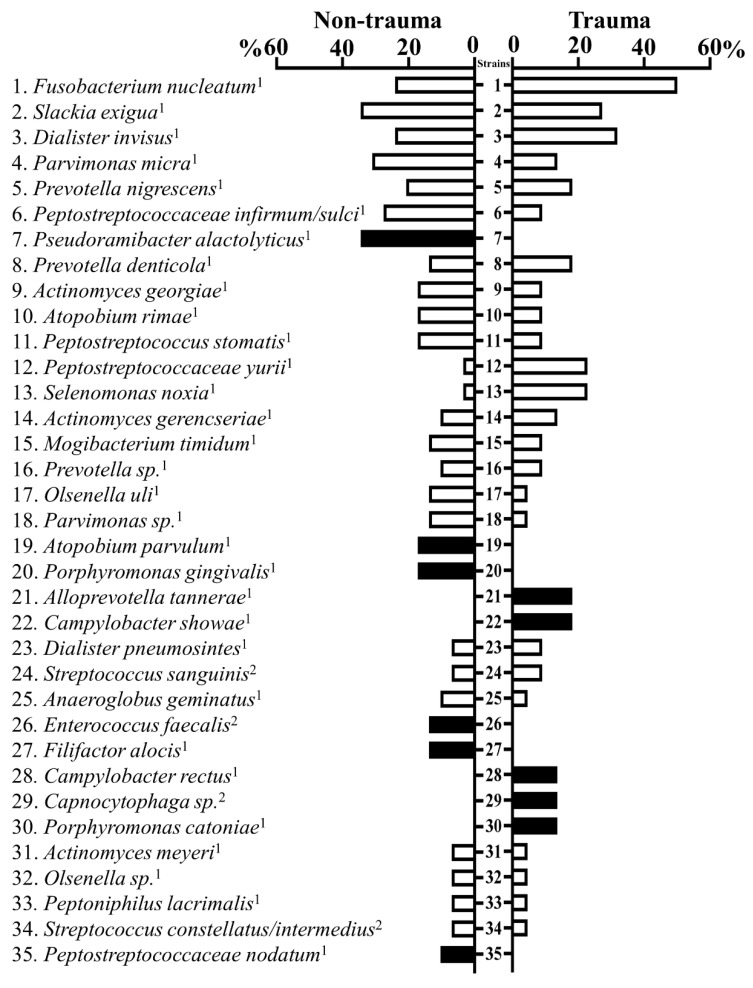
*Fusobacterium nucleatum* and *Slackia exigua* were the most prevalent species from root canals identified by culture. The bar chart shows percentages of teeth (trauma and non-trauma) containing the most common species identified by 16S rDNA Polymerase Chain Reaction sequenced. Anaerobic species are marked by the superscript “1” and species with respiratory metabolism (aerobic, microaerophilic, and facultative anaerobic) are marked by the superscript “2”.

**Figure 3 jcm-09-03877-f003:**
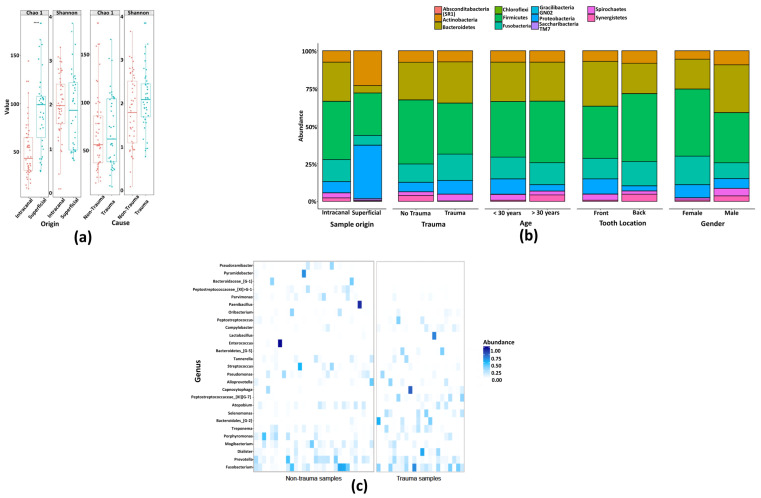
Illumina MiSeq sequencing showed an overall high microbial diversity. (**a**) Boxplot of alpha-diversity indices indicate a great richness of the studied bacterial communities. (**b**) Taxonomical composition of phyla of bacterial communities in root canal samples grouped by sample origin, history of trauma, age, tooth location, and gender. (**c**) Root canal community profile at genus level based on bacterial prevalence.

**Figure 4 jcm-09-03877-f004:**
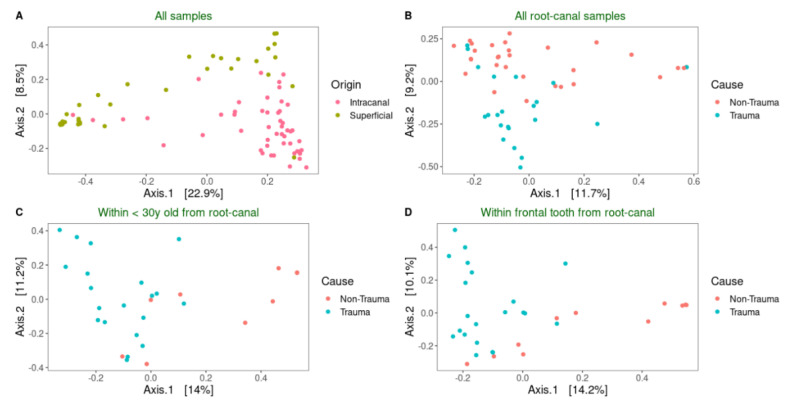
The bacterial profile clustered samples differently based on trauma history, age, and tooth position. Principal coordinate analysis (PCoA) score plots of bacterial community composition based on Bray-Curtis similarities of the relative abundances of HMTs. Beta diversity was calculated for all samples according to (**A**) origin of the sample (tooth surface and root canal), (**B**) trauma history (non-trauma and trauma), (**C**) within <30y root canal samples, and (**D**) within root canal samples with frontal tooth position.

**Figure 5 jcm-09-03877-f005:**
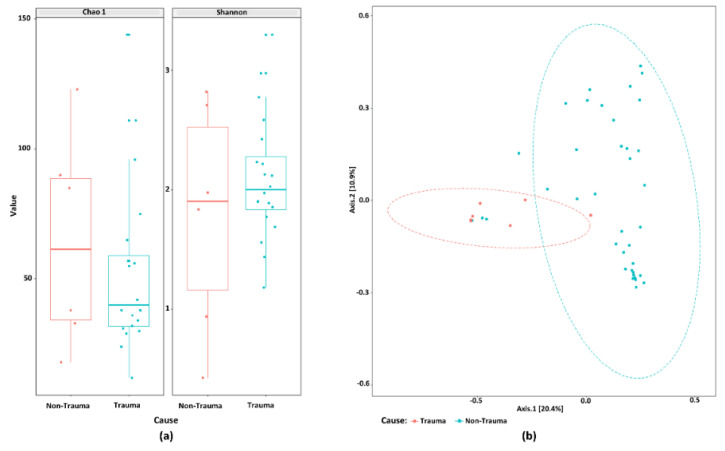
The microbiome composition from root canal samples that are frontal and below 30-years-old discriminates traumatized teeth from other primary infected non-traumatized teeth. (**a**) Boxplot of alpha-diversity indices of the studied bacterial communities in these samples. (**b**) Principal coordinate analysis (PCoA) score plots of bacterial community composition when age and tooth position are considered in the analysis. Ellipses drawn based on the multivariate t-distribution of the samples.

**Figure 6 jcm-09-03877-f006:**
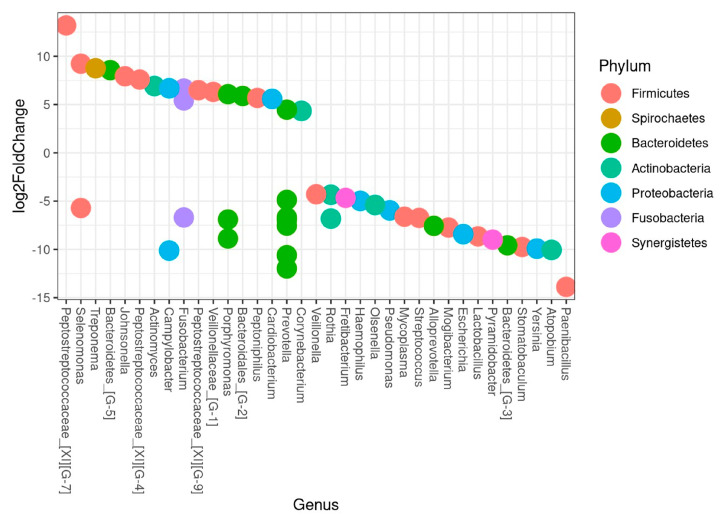
Being in the trauma group was significantly associated with species playing a crucial role in biofilm formation within samples from frontal teeth of subjects below 30-years-old. Bacterial association to trauma and non-trauma teeth presented at genus level. HMTs with adjusted *p*-values < 0.01 are plotted in the figure. Each circle represents an HMT. If the log2FoldChange is above ZERO the HMTs were more abundant in trauma samples, and below ZERO, the HMTs were more abundant in non-trauma samples.

**Table 1 jcm-09-03877-t001:** Description of samples characteristics by trauma association.

	Trauma (*n* = 30)	Non-Trauma (*n* = 32)	*p*-Value
Age in years, mean	16.6	50.0	<0.001
Gender, % female	63.0	58.1	0.110
Tooth position, % front	96	35.5	<0.001
Presence of sinus tract, % positive	29.6	35.5	0.781
Pain, % positive	11.1	22.6	0.311
CFU viable bacteria ^1^	359090	1119413	0.107

^1^ Colony-forming units (CFU). Difference in group means were tested with analysis of variance (ANOVA) for continuous variables, and by chi-square test for categorical variables.

**Table 2 jcm-09-03877-t002:** Overview of Illumina MiSeq sequencing by bioinformatics processing.

	Illumina MiSeq					
	Surface Control			Root Canal Samples		
	Trauma	Non-Trauma	All	Trauma	Non-Trauma	All
No. of samples	19	19	38	22	30	52
Sequence length (bp)	2 × 250	2 × 250	2 × 250	2 × 250	2 × 250	2 × 250
Total number of reads after filtering (in millions)	~1.9	~2.3	~4.2	~2.8	~3.5	~6.3
Reads per sample, mean (min-max) (in thousands)	~97 (39–189)	~125 (64–181)	~111 (39–189)	~128 (54–213)	~114 (37–295)	~120 (37–295)
Phyla	10	11	11	10	9	11
Genera	151	134	162	134	120	150
Species/HMT ^1^	399	441	502	353	368	454

^1^ Represents the number of human microbial taxons (HMT) in all samples according to each category. HMT is a given phylotype defined by using a 98.5% sequence similarity cut-off.

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
