# Peer review of "New Insights into the Microbial Profiles of Infected Root Canals in Traumatized Teeth"

_jcm, 2020, doi:10.3390/jcm9123877_

Round 1
Reviewer 1 Report
The authors took the sensible approach to investigate the matter.
The material obtained from the root canals was secured for the field contamination which is of great importance.
The population of the study was not homogenous regarding the age and tooth position. As far as the age is strictly related to the age pattern of dentoalveolar trauma the control group might be corrected for the position of the tooth as it may be of importance (more complex canal system). However this is of minor importance for the whole research.
Reviewer 2 Report
The use of current methods of characterisation of dental microbiotes is interesting and makes it possible to describe and discriminate between the two groups of traumatised and non-traumatised teeth. However, these data remain strictly descriptive.The search for a difference between these two groups is poorly justified, in particular because the pulpal diagnoses, the routes of contamination and the dates of contamination are not specified. In this situation, the clinical implications cannot be advanced nor discussed. The search for a difference between these two groups is poorly justified, in particular because the pulpal diagnoses, the routes of contamination and the dates of contamination are not specified. In this situation, the clinical implications cannot be advanced or discussed.
How can the observed differences be clarified, what are the obstacles to interpreting the results? For clinicians who are not biologists, it would be interesting to differentiate between aerobic and anaerobic strains. What hypotheses could be put forward to explain these differences?
